# Estimation of Shear Modulus and Hardness of High-Entropy Alloys Made from Early Transition Metals Based on Bonding Parameters

**DOI:** 10.3390/ma16062311

**Published:** 2023-03-13

**Authors:** Ottó Temesi, Lajos K. Varga, Xiaoqing Li, Levente Vitos, Nguyen Q. Chinh

**Affiliations:** 1H-ION Research, Development and Innovation Ltd., Konkoly-Thege Miklos út 29-33, H-1121 Budapest, Hungary; 2Department of Materials Physics, Eötvös Loránd University, Pázmány Péter Sétány 1/A, H-1117 Budapest, Hungary; 3Wigner Research Centre for Physics, Institute for Solid State Physics and Optics, Konkoly-Thege Miklos út 29-33, H-1121 Budapest, Hungary; 4Applied Materials Physics, Department of Materials Science and Engineering, Royal Institute of Technology, SE-100 44 Stockholm, Sweden

**Keywords:** Early Transition Metals (ETM), Refractory High Entropy Alloy (RHEA), shear modulus (G), valence electron concentration (*VEC*), enthalpy of mixing (Δ*H_mix_*), Vickers hardness (*HV*)

## Abstract

The relationship between the tendencies towards rigidity (measured by shear modulus, *G*) and hardness (measured by Vickers hardness, *HV*) of early transition metal (ETM)-based refractory high-entropy alloys (RHEA) and bond parameters (i.e., valence electron concentration (*VEC*), enthalpy of mixing (Δ*H_mix_*)) was investigated. These bond parameters, *VEC* and Δ*H_mix_*, are available from composition and tabulated data, respectively. Based on our own data (9 samples) and those available from the literatures (47 + 27 samples), it seems that for ETM-based RHEAs the *G* and *HV* characteristics have a close correlation with the bonding parameters. The room temperature value of *G* and *HV* increases with the *VEC* and with the negative value of Δ*H_mix_*. Corresponding equations were deduced for the first time through multiple linear regression analysis, in order to help design the mechanical properties of ETM refractory high-entropy alloys.

## 1. Introduction

The name high-entropy alloy (HEA) originated from Yeh et al. [1] almost 20 years ago. This name came from the role of the high mixing entropy, as the dominant term of the Gibbs free energy described by the equation $\;\Delta G_{mix}=\Delta H_{mix}-T\Delta S_{mix}$, where $\Delta H_{mix}\;$ and $\Delta S_{mix}\;$ are the enthalpy and entropy of mixing, respectively, and *T* is the absolute temperature. In the last two decades, the study of high-entropy alloys (HEAs) has been at the forefront of materials science, due to their unique mechanical [2,3,4,5,6] and corrosion resistive [7,8,9] behaviors.

Early transition metal (ETM)-based refractory high-entropy alloys (RHEAs) containing only a single-phase body-centered cubic (BCC) structure have been intensively studied since the late 2000s [10]. In addition to their excellent mechanical properties, the RHEAs suffer from two main drawbacks: a lack of oxidation resistance when working in air at temperatures above 1000 K, and a poor ductility at room temperature in the as-cast state. In the last 10 years, many criteria for the ductile–brittle behavior of these materials have been presented [11]. It should be emphasized that all of them contain the shear modulus, *G*.

It is well known that shear modulus characterizes the rigidity of a sample and it is a measure of the resistance to shape change. A cube can be deformed by stretching along the diagonal of one of the faces (trigonal distortion) and along the axis (tetragonal distortion). The relevant elastic constants in these cases are called *C_44_* and *C′*, respectively. The shear modulus can be expressed as a function of these elastic constants [12] as
(1)$$G_{V}=C_{44}-\frac{2}{5}\left(C_{44}-C’\right)$$
in Voigt’s notation [12] and
(2)$$G_{R}=\frac{5C_{44}C’}{2C_{44}+3C’}\;$$
in Reuss’s notation [12], where
(3)$$C’=\frac{C_{11}-C_{12}}{2}$$

The *G* will be the arithmetic Hill average of the lower (*G_V_*) and upper (*G_R_*) limits:(4)$$G=\frac{G_{V}+G_{R}}{2}\;$$

It should be mentioned that for an isotropic cubic alloy *C_44_* = *C′*, and in this case all of elastic constants are the same, that is
*G_V_ = G_R_ = G = C_44_ = C′*(5)

In general, the deviation from isotropy can be measured, for example, by using Zener’s anisotropy, *A_Z_* defined as [12]:(6)$$A_{Z}=\frac{C_{44}}{C’}$$
which is equal to “1” for perfect isotropy and can be as high as “10” for amorphous alloys [12].

In order to help in designing ductile RHEAs, here we propose an empirical criterion for determining the rigidity *G* and hardness *HV* by means of bond parametric functions, which comprise of *VEC* and *ΔH_mix_*, which are important characteristics of high-entropy alloys [13]. These parameters are defined using well-known formulas:(7)$$VEC=\sum_{i}^{n}VEC_{i}\cdot c_{i}$$
where *VEC_i_* is the valence electron concentration of element *i* with atomic concentration *c_i_*, and
(8)$$\Delta H=\sum_{i<j}4H_{ij}c_{i}c_{j}$$
where *H_ij_* is the enthalpy of mixing of elements *i* and *j* at the equimolar concentration in regular binary solutions [13].

## 2. Materials and Methods

Metallic elements in wire and chunk form, having a purity of 99.95%, were used to produce samples of 15 g. The elongated ellipsoid shape rods about 40 mm long and with a diameter of about 10 mm were prepared by induction melting in a water-cooled copper mold under argon atmosphere. The rods were re-melted 5 times and held above the melting point to homogenize the ingot. Nine different composition samples (see Table 1) were prepared. All of these 9 samples had a single phase BCC structure, which was confirmed by XRD investigations.

Microhardness data (*HV*) were determined at room temperature on the mirror-polished surface perpendicular to the axis of the sample rod. The hardness measurements were carried out using a Vickers type indenter at a load of 1 kg on a Zwick/Roell-ZHμ-Indentec microhardness tester. At least ten measurements were performed on each sample, and then their average was taken as the characteristic value for the sample.

The elastic constancies *C_11_* and *C_12_* were determined from the bulk modulus, *B,* using the equation in [16]:*B = (C_11_+2C_12_)/3*
and from the tetragonal component, *C*′, of the shear modulus described in Equation (3).

In the ab initio calculations, the bulk modulus was extracted from the Morse function fitted to the total energies, calculated as a function of volume. The total energy and the two components of the shear elastic parameters, *C′* and *C_44_*, were computed according to the standard methodology of density functional theory. More details of the process can be found in Ref. [22].

## 3. Results and Discussion

### 3.1. Estimation of Shear Modulus, G, Based on Bonding Parameters

In a seminal paper [23], Saito et al. showed 20 years ago that the *C′* component of *G varies* linearly with the *VEC* number, and it becomes zero (that is *C_11_ = C_12_*) around *VEC* = 4.2 for a set of Ti-X binary BCC alloys, where X may be Nb, Ta, V, or Mo element. For a Ti-based crystalline alloy (Ti-23Nb-0.7Ta-2Zr-1.2O), the *VEC* was tuned to the “magic” 4.2 value by alloying with oxygen.

The basic assumption of the present work was that the mentioned linearity also applies to RHEAs, which are constituted from a mixture of ETM elements. In the present work, we considered the following ETM elements, with the *VEC* between 3 and 6 written in parentheses after the element: Yttrium (3), Titanium (4), Zirconium (4), Hafnium (4), Vanadium (5), Niobium (5), Tantalum (5), Chromium (6), Molybdenum (6), and Tungsten (6).

In order to confirm the mentioned assumption, elastic constants of several samples containing ETM elements were determined using ab initio calculations. These values and those from the literature are listed in Table 1. Our conjecture is confirmed on Figure 1, where we have collected the elastic constant, *C′*, data for all the ETM-RHEA’s samples available in the literature having single-phase BCC structure, as well as the data of the nine samples prepared for the present work.

Figure 1 shows the values of *C′* (in Figure 1a) and *C_44_* (in Figure 1b) as a function of the *VEC* number. It can be very clearly seen that the parameter C′ visibly changed linearly with the *VEC* number, completely independently of the composition of the samples. This means that there is a good correlation between the *C′* component of the *G* and the *VEC* number. On the contrary, a correlation cannot be observed between the *C_44_* component and the *VEC* number.

Figure 2 shows the relationships between *G* and *VEC* (Figure 2a), as well as between *G* and Δ*H_mix_* (Figure 2b). It can be seen that none of them showed acceptable R^2^ values of correlation for a linear fitting. However, the analysis showed that a good multiple linear regression can be used for fitting *G* as a function of *VEC* and Δ*H_mix_* in the form:*G = a_o_ + a_1_**× VEC + a_2_**× ΔH_mix_*(9)
where *a_o_* is constant, and *a_1_* and *a_2_* are proportional coefficients, which can be obtained using multiple linear regression for the data listed in Table 1.

As a result, the *G* for the RHEAs system can be given as *G_fitted_*, where:*G_fitted_ =* −110.68 + 34.87 × *VEC* + 0.73 × Δ*H_mix_*(10)

According to Equation (10), the *G* of any ETM-based RHEA can be estimated using the *VEC* number and Δ*H_mix_* mixing enthalpy calculated with Equations (7) and (8), respectively.

The values of *G_fitted_* are also listed in Table 1, together with the accepted ones. Figure 3 shows the correlation between the accepted and fitted values of *G*, indicating that the proposed model is rational. It can clearly be seen that a linear proportionality with a slope of 1 (function of type f(x) = x) can be fitted to the data, clearly confirming the validity of Equation (10) for an ETM-based RHEA system.

Furthermore, from Equation (10) we have:*VEC* = −(0.73/34.87) × Δ*H_mix_* + (*G* + 110.68)/34.87(11)
which means that at a given value of *G*, the *VEC* parameter changes linearly with the mixing enthalpy, Δ*H_mix_*. A set of straight line *VEC* versus Δ*H_mix_* obtained at different values of G can be seen in Figure 4.

Figure 4 illustrates the relationship between *G* and bonding parameters. The variable range of *G* caused by *VEC* is 20–100 GPa, with *VEC* increasing from 3.5 to 6.5. It seems that the effect of the average valence electron concentration is more significant than that of the enthalpy of mixing in determining the rigidity of RHEAs.

### 3.2. Estimation of HV, Based on Bonding Parameters

In Table 2, the *HV* of 36 samples, including those from the present work and the literature, is listed, together with the *VEC* numbers and *ΔH_mix_* calculated using Equations (7) and (8).

Using the data in Table 2, Figure 5 shows the relationships between *HV* and *VEC* (Figure 5a), as well as between *HV* and Δ*H_mix_* (Figure 5b).

Based on the correlations shown in Figure 5, it can be assumed that the *HV* can also be expressed using a linear combination of the two bonding parameters, *VEC* and Δ*H_mix_*. Appling the multiple linear regression, *HV* can be given by the following formula:*HV_fitted_* = −122.18 + 109.75 × *VEC* − 11.23 × Δ*H_mix_*(12)

According to Equation (12), the *HV* of any ETM-based RHEA can be estimated using the *VEC* number and Δ*H_mix_* mixing enthalpy calculated by Equations (7) and (8), respectively. This estimated *HV* is represented in Figure 6 as a function of the measured one. It can be seen that the data are gathered around a bisector, indicating the good correlation between the estimated and measured values.

It is important to note that taking the upper limiting values for the bond parameters, at *VEC* = 6 and Δ*H_mi_*_x_= −10 kJ/mol, the maximal value of *HV* can be predicted as:*HV_max_* = −122.18 + 109.75 × 6−11.23 × (−10) = 649 kgf/mm^2^(13)
for an ETM-based RHEA system. This *HV* value (13) is 6490 MPa. Considering the yield stress, $\sigma_{Y}$ as one-third value of the *HV* [27], that is:(14)$$\sigma_{Y}\approx HV/3,$$

The maximum yield stress of an ETM-based RHEA system can be estimated to be about 2100–2200 MPa.

Looking at Equation (12), it is clear that the *HV* of RHEAs increases with an increasing *VEC* number and, as well as with the negative Δ*H_mix_*. Furthermore, from Equation (12) we have:*VEC* = (11.23/109.75) × Δ*H_mix_ + (HV* + 122.18)/109*.75*,(15)
that is, at a given value of *HV*, the *VEC* parameter changes linearly with the Δ*H_mix_*. A set of straight line *VEC* versus Δ*H_mix_* obtained at different values of *HV* can be seen in Figure 7.

Figure 7 illustrates the relationship between the hardness and bonding parameters. The variable range of *HV* caused by *VEC* is 200–600 kgf/mm^2^ with *VEC* increasing from 3.6 to 6.4. It seems that, similarly to in the case of shear modulus, the effect of average valence electron concentration is more significant than that of the enthalpy of mixing in determining the hardness of RHEAs.

## 4. Conclusions

Our results showed that the shear modulus, *G,* and *HV* characteristics could be correlated with two easy to determine bond parameters, *VEC* and Δ*H_mix_*. These bond parameters can be obtained using tabulated data of the elements and composition of the alloy. The correlation of bond parameters with the accepted values of *G* and *HV* was demonstrated using multiple linear regression calculations. The accepted values of *G* were obtained from elastic constants *C′* and *C_44_* and those of *HV* were determined using experimental measurements. Considering the limits of variable range for *VEC* (3.75 and 6) and for Δ*H_mix_*(−10 and 8 kJ/mol), for the ETM-based HEAs, the maximal hardness that can be foreseen for a single-phase BCC structure is about 649 kgf/mm^2^ (6490 MPa). Taking into account the correlation between the hardness and yield stress (see Equation (14)), the maximal yield stress is expected to be around 2170 MPa. It should also be emphasized that it is possible to adapt the developed equations for all of ETM refractory high-entropy alloys, with different compositions obtainable by combination of the nine refractory elements.

It is of importance to note that the relationship between *G* (and *HV*) and the bond parameters can probably be applied to late transition metal-based HEAs as well. The corresponding proportional coefficients will be published at a later date.

## Figures and Tables

**Figure 1 materials-16-02311-f001:**
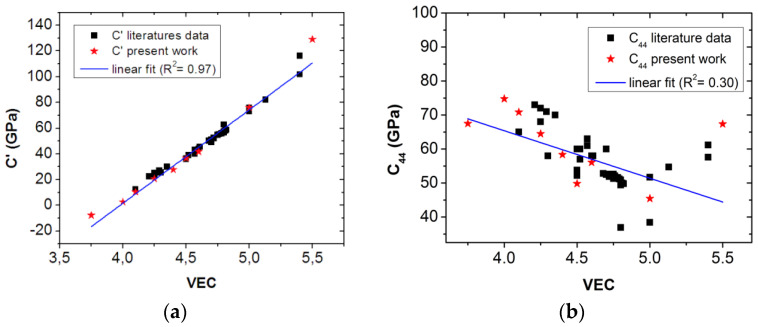
The C′ component of *G* changes linearly (**a**), whereas *C_44_* does not correlate (**b**) with the *VEC*.

**Figure 2 materials-16-02311-f002:**
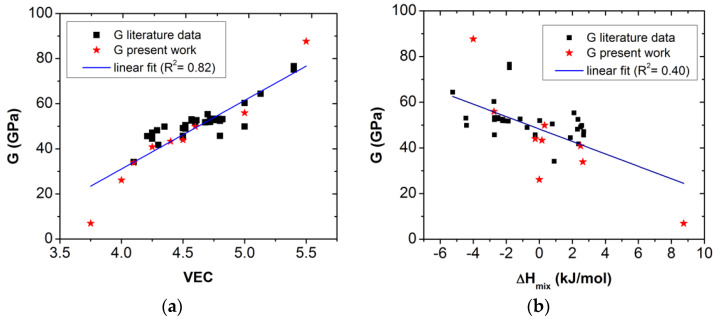
*G*, in the function of *VEC* (**a**); and of Δ*H_mix_* (**b**) for the various ETM-based RHEA samples listed in Table 1.

**Figure 3 materials-16-02311-f003:**
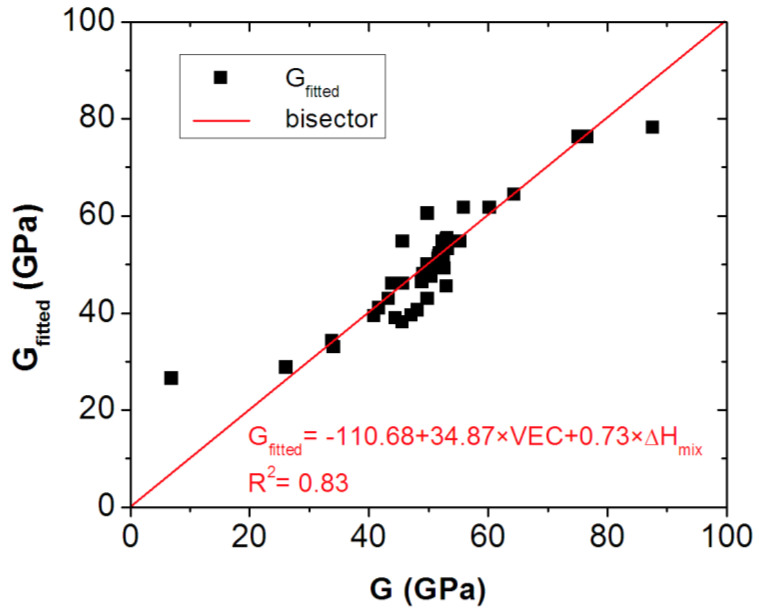
The accepted and fitted values of G line up around the first bisector, the fitted value can be calculated as G = −110.68 + 34.87 × VEC + 0.73 × Δ*H_mix_* with a goodness of fit R^2^ value of 0.83.

**Figure 4 materials-16-02311-f004:**
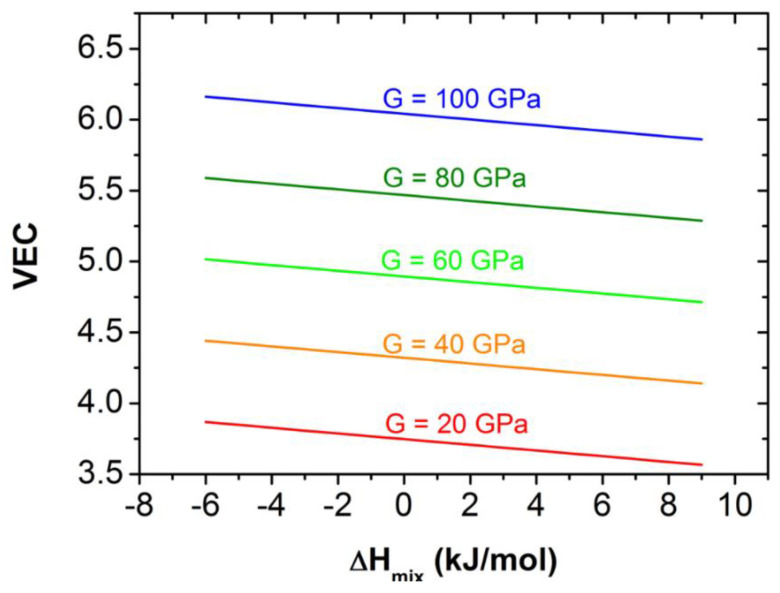
Relationship between *G* and bond parameters for ETM–RHEAs, indicated by linear *VEC*-Δ*H_mix_* functions obtained at different *G* values.

**Figure 5 materials-16-02311-f005:**
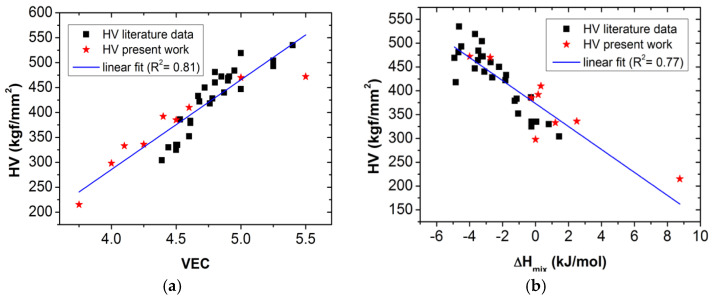
*HV*, in the function of VEC (**a**); and of Δ*H_mix_* (**b**) for the various ETM-based RHEA samples listed in Table 2.

**Figure 6 materials-16-02311-f006:**
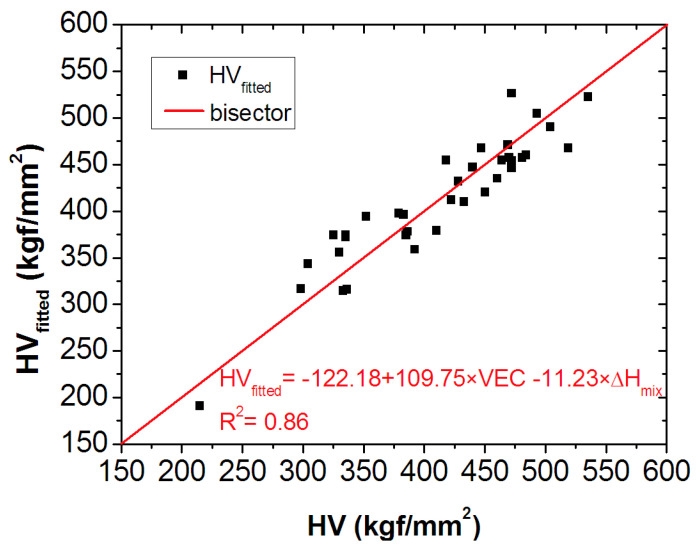
Relationship between the accepted and fitted values of *HV*, which line up around the first bisector.

**Figure 7 materials-16-02311-f007:**
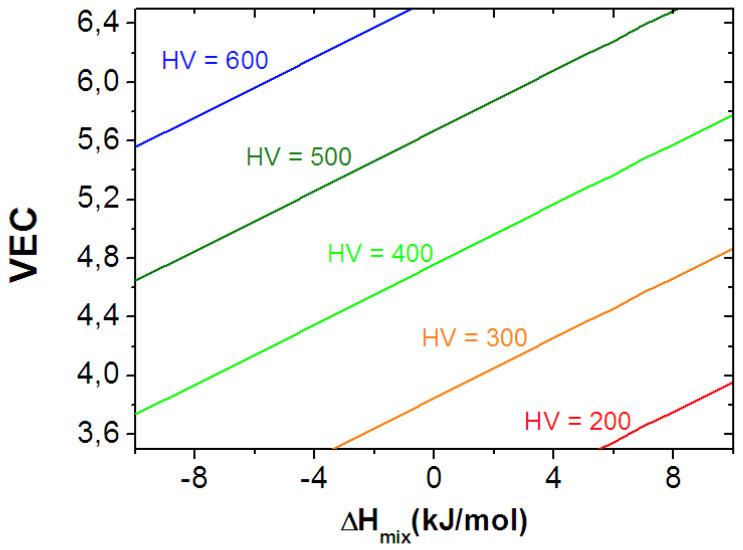
Relationship between *HV* and bond parameters for ETM–RHEAs, indicated by linear *VEC-* Δ*H_mix_* functions obtained at different *HV* values.

**Table 1 materials-16-02311-t001:** Ab initio calculated elastic constants (in GPa), valence electron concentration, enthalpy of mixing (in kJ/mole), and shear modulus (in GPa) for the samples of the present work (pw) and those taken from the literature.

Alloys	Ref	VEC	C_11_	C_12_	C_44_	C′	Δ*H_mix_*	G	G_fitted_
Y25Ti25Zr25Hf25	pw	3.75	70.40	85.94	67.48	−7.77	8.75	6.95	26.21
Ti33.33Zr33.33Hf33.34	pw	4.00	99.99	94.67	74.76	2.66	0.00	26.12	28.80
Ti45Zr45Nb5Ta5	[14]	4.10	112.00	87.00	65.00	12.50	0.90	34.13	32.92
Ti30Zr30Hf30Nb10	pw	4.10	114.88	94.34	70.83	10.27	2.64	33.85	34.14
Ti26.3Zr26.3Hf26.3Nb10.55Ta10.55	[14]	4.21	145.00	100.00	73.00	22.50	2.68	45.63	38.00
Ti25Zr25Hf25Nb12.5Ta12.5	[14]	4.25	151.00	101.00	72.00	25.00	2.69	47.15	39.40
Ti37.5Zr25Ta12.5Hf12.5Nb12.5	[14]	4.25	146.00	99.00	68.00	23.50	1.88	44.45	38.83
Ti25Zr25Hf25Nb25	pw	4.25	136.20	95.06	64.48	20.57	2.50	40.85	39.27
Ti23.8Zr23.8Hf23.8Nb14.3Ta14.3	[14]	4.29	156.00	102.00	71.00	27.00	2.31	48.19	40.53
Ti35Zr35Nb25Ta5	[14]	4.30	142.00	91.00	58.00	25.50	2.38	41.71	40.93
Ti21.7Zr21.7Hf21.7Nb17.45Ta17.45	[14]	4.35	164.00	104.00	70.00	30.00	2.57	49.83	42.80
Ti20Zr20Hf20Nb20V20	pw	4.40	149.70	94.20	58.35	27.75	0.16	43.30	42.86
NbTiVZr	[15]	4.50	166.10	93.80	52.20	36.15	−0.25	45.05	46.06
NbTiVZr	[16]	4.50	166.40	94.70	53.80	35.85	−0.25	45.72	46.06
Ti25Nb25Ta25Zr25	[14]	4.50	174.00	101.00	60.00	36.50	2.50	49.16	47.99
Ti25Zr25V25Nb25	pw	4.50	165.26	92.22	49.80	36.52	−0.25	43.98	46.06
Ti25Nb25Ta25Zr25	[14]	4.50	174.00	101.00	60.00	36.50	2.50	49.16	47.99
Ti30.5Zr30.5Nb13Ta13Mo13	[14]	4.52	175.00	97.00	57.00	39.00	−0.74	48.96	46.41
Ti34Zr20Nb20Ta20Mo6	[14]	4.52	180.00	102.00	60.00	39.00	0.79	50.48	47.49
Ti21.67Zr21.67Nb21.66Ta35	[14]	4.57	187.00	107.00	63.00	40.00	2.34	52.51	50.31
Ti23.8Zr23.8Hf23.8Cr4.8Mo23.8	[14]	4.57	192.00	106.00	61.00	43.00	−4.45	53.03	45.56
Ti30Zr20Nb20Ta20Mo10	[14]	4.60	192.00	104.00	58.00	44.00	0.02	51.93	49.74
Ti20Zr20V20Nb20Ta20	pw	4.60	185.98	102.36	56.08	41.81	0.32	49.86	49.95
Ti39.4Nb15.15Ta15.15Zr15.15Mo15.15	[14]	4.61	195.00	104.00	58.00	45.50	−1.16	52.63	49.26
Mo0.8NbTiZr	[16]	4.68	199.00	98.70	52.80	50.15	-1.88	51.72	51.20
TiZrNbMo0.8	[17]	4.68	199.00	98.70	52.80	50.15	−1.88	51.72	51.20
Mo0.8NbTiV0.2Zr	[16]	4.70	200.80	99.00	52.50	50.90	−2.05	51.85	51.77
Ti15Zr15Nb35Ta35	[14]	4.70	210.00	112.00	60.00	49.00	2.10	55.33	54.68
TiZrNbMo0.8V0.2	[17]	4.70	200.80	99.00	52.50	50.90	−2.05	51.85	51.77
Mo0.9NbTiZr	[16]	4.72	204.30	99.50	52.60	52.40	−2.21	52.52	52.36
TiZrNbMo0.9	[17]	4.72	204.30	99.50	52.60	52.40	−2.21	52.52	52.36
Mo0.8NbTiV0.5Zr	[16]	4.72	203.70	100.00	51.90	51.85	−2.23	51.88	52.35
TiZrNbMo0.8V0.5	[17]	4.72	203.70	100.00	51.90	51.85	−2.23	51.88	52.35
MoNbTiZr	[15]	4.75	209.80	99.90	51.30	54.95	−2.50	52.73	53.20
MoNbTiZr	[16]	4.75	209.90	101.00	52.60	54.45	−2.50	53.33	53.20
TiZrNbMo	[17]	4.75	209.90	101.00	52.60	54.45	−2.50	53.33	53.20
MoNbTiV0.25Zr	[16]	4.76	211.00	100.60	52.10	55.20	−2.60	53.32	53.48
TiZrNbMoV0.25	[17]	4.76	211.00	100.60	52.10	55.20	−2.60	53.32	53.48
MoNbTiV0.5Zr	[16]	4.78	212.20	100.30	51.60	55.95	−2.67	53.30	54.13
TiZrNbMoV0.50	[17]	4.78	212.20	100.30	51.60	55.95	-2.67	53.30	54.13
MoNbTiV0.75Zr	[16]	4.79	213.20	100.30	51.20	56.45	−2.70	53.24	54.46
TiZrNbMoV0.75	[17]	4.79	213.20	100.30	51.20	56.45	−2.70	53.24	54.46
MoNbTiVZr	[16]	4.80	213.70	100.70	50.90	56.50	−2.72	53.07	54.79
MoNbTiVZr	[15]	4.80	215.00	100.50	49.40	57.25	−2.72	52.40	54.79
MoNbTiVZr	[18]	4.80	231.00	105.60	36.90	62.70	−2.72	45.70	54.79
TiZrNbMoV1.00	[17]	4.80	213.70	100.70	50.90	56.50	−2.72	53.07	54.79
MoNbTiV1.25Zr	[16]	4.81	218.00	101.90	50.00	58.05	−2.72	53.08	55.14
TiZrNbMoV1.25	[17]	4.81	218.00	101.90	50.00	58.05	−2.72	53.08	55.14
MoNbTiV1.5Zr	[16]	4.82	219.30	102.20	49.80	58.55	−2.71	53.13	55.50
TiZrNbMoV1.50	[17]	4.82	219.30	102.20	49.80	58.55	−2.71	53.13	55.50
CrMoNbTaTiVZr	[19]	5.00	261.40	115.10	38.40	73.15	−4.41	49.85	60.58
MoNbTiV	[20]	5.00	265.50	114.00	51.70	75.75	−2.75	60.27	61.75
Ti25V25Nb25Mo25	pw	5.00	264.13	112.11	45.45	76.01	−2.75	55.92	61.75
CrMoNbTaTiVWZr	[19]	5.13	286.10	121.90	54.70	82.10	−5.25	64.39	64.53
MoNbTaVW	[18]	5.40	392.70	160.10	57.60	116.30	−1.81	76.63	76.35
MoNbTaVW	[21]	5.40	380.80	177.30	61.20	101.75	−1.81	75.11	76.35
V25Nb25Mo25W25	pw	5.50	390.15	131.97	67.35	129.09	−4.00	87.66	78.31

**Table 2 materials-16-02311-t002:** *HV* data (kgf/mm^2^) together with the *VEC* and Δ*H_mix_* (kJ/mole) for the samples of the present work and those taken from the literature.

Alloys	Ref	VEC	HV	ΔH_mix_	HV _Fitted_
Nb28.3Ti24.5V23Zr24.2	[24]	4.51	335	0.05	376
Nb22.6Ti19.4V37.2Zr20.8	[24]	4.60	352	−1.05	397
Cr24.6Nb26.7Ti23.9Zr24.8	[24]	4.76	418	−4.84	451
Cr20.2Nb20Ti19.9V19.6Zr20.3	[24]	4.80	481	−4.68	454
Y25Ti25Zr25Hf25	pw	3.75	215	8.75	204
Ti33.33Zr33.33Hf33.34	pw	4.00	298	0.00	315
Ti30Zr30Hf30Nb10	pw	4.10	333	1.20	316
Ti25Zr25Hf25Nb25	pw	4.25	336	2.50	322
Ti20Zr20Hf20Nb20V20	pw	4.40	392	0.16	362
Ti25Zr25V25Nb25	pw	4.50	385	−0.25	377
Ti20Zr20V20Nb20Ta20	pw	4.60	410	0.32	384
Ti25V25Nb25Mo25	pw	5.00	470	−2.75	461
V25Nb25Mo25W25	pw	5.50	472	−4.00	532
TiZrNbV	[25]	4.50	325	−0.25	377
TiZrNbVMo0.3	[25]	4.61	379	−1.26	399
TiZrNbVMo0.5	[25]	4.67	433	−1.78	412
TiZrNbVMo0.7	[25]	4.72	450	−2.21	422
TiZrNbVMo1.0	[25]	4.80	460	−2.72	436
TiZrNbVMo1.3.	[25]	4.87	440	−3.10	448
TiZrNbVMo1.5	[25]	4.91	472	−3.31	455
TiZrNbVMo1.7	[25]	4.95	484	−3.47	461
TiZrNbVMo2.0	[25]	5.00	519	−3.67	469
TiZrNbV0.3	[25]	4.39	304	1.43	349
TiZrNbV0.3Mo0.1	[25]	4.44	330	0.80	361
TiZrNbV0.3Mo0.3	[25]	4.53	386	−0.28	381
TiZrNbV0.3Mo0.5	[25]	4.61	383	−1.14	398
TiZrNbV0.3Mo0.7	[25]	4.68	422	−1.83	413
TiZrNbV0.Mo1.03	[25]	4.78	428	−2.62	432
TiZrNbV0.3Mo1.3	[25]	4.85	472	−3.20	447
TiZrNbV0.3Mo1.5	[25]	4.90	464	−3.49	455
NbCrMo0.5Ta0,5TiZr	[25]	4.90	469	−4.92	469
MoNbTaV	[26]	5.25	504	−3.25	495
NbTaVW	[26]	5.25	493	−4.50	507
NbTaTiVW	[26]	5.00	447	−3.68	469
MoNbTaVW	[26]	5.40	535	−4.64	526
NbTiVZr	[26]	4.50	335	−0.25	377

## Data Availability

The data that support the findings of this study are partly taken from the cited references and are partly the results of the authors and not available elsewhere.

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
