# Peer review of "Estimation of Shear Modulus and Hardness of High-Entropy Alloys Made from Early Transition Metals Based on Bonding Parameters"

_materials, 2023, doi:10.3390/ma16062311_

Round 1

Reviewer 1 Report

The manuscript in its current state could be accepted after major revision.

Before accepting the manuscript in its current state, some aspects should be corrected.

1.- In line 65, it is specified that the hardness test was carried out at room temperature; it is not necessary to repeat it in line 67

2.- The quality of the presentation should be better. The order of the tables/figures and text should be rearranged since it is complicated to read like this. First, the description and discussion of the figure/table must be mentioned, and then they must be presented.

3.- If a word or term was already abbreviated (through letters or symbols) for the first time within the text, it should not be repeated and only use the abbreviation or symbol. The same applies to the caption of figures or tables.

4.- In line 125, the word Fitted must be lowercase.

5.- Bibliographical references should be added indicating the limit for considering that the R2 value has or does not have a reasonable correlation in linear fit.

6.- The "rule of thumb" is only mentioned in the conclusions; it should be added in the manuscript and referenced; in addition also should is referenced the yield strength value of the "best Steel" e indicated what is.

7.- In the references section, these must be annotated homogeneously.

Reviewer 2 Report

Comments and suggestions on this work can be found in the attached file.

Round 2

Reviewer 1 Report

The manuscript in its current state could be accepted after major revision.

Before accepting the manuscript in its current state, some aspects should be corrected.

1.- The quality of the presentation should be better. The order of the tables/figures and text should be rearranged since it is complicated to read like this. First, the description and discussion of the figure/table must be mentioned, and then they must be presented.

2.- If a word or term was already abbreviated (through letters or symbols) for the first time within the text, it should not be repeated and only use the abbreviation or symbol. The same applies to the caption of figures or tables.

Lines 59, 103, 122, 134, 141,142, 150, 156, 178,179 and shear modulus (G)

3.- The equation of line 76 needs to be enumerated.

4.- Regarding the following statement that only appears in the conclusions "rule of thumb concerning the correlation between the hardness and yield stress, the maximal yield stress is expected around 2170 MPa (as a reminder, the best steel has a yield stress around 1500 MPa)". First, the rule of thumb concerning the correlation between hardness and yield stress should be described and referenced in the manuscript. Second, it should be added and referenced in the manuscript what is the "best Steel."
